# Value Assessment of Health Losses Caused by PM_2.5_ in Changsha City, China

**DOI:** 10.3390/ijerph16112063

**Published:** 2019-06-11

**Authors:** Guanghui Yu, Feifan Wang, Jing Hu, Yan Liao, Xianzhao Liu

**Affiliations:** 1The School of Resource, Environment and Safety Engineering, Hunan University of Science and Technology, Xiangtan 411201, China; wff390@163.com (F.W.); hujing081@163.com (J.H.); 15197292263@163.com (X.L.); 2South China Institute of Environmental Science, Ministry of Ecology and Environment (MEE), Guangzhou 510655, China; liaoyan@scies.org

**Keywords:** PM_2.5_, exposure–response coefficient, health economic loss, Changsha

## Abstract

With the advancement of urbanization, the harm caused to human health by PM_2.5_ pollution has been receiving increasing attention worldwide. In order to increase public awareness and understanding of the damage caused by PM_2.5_ in the air and gain the attention of relevant management departments, Changsha City is used as the research object, and the environmental quality data and public health data of Changsha City from 2013 to 2017 are used. All-cause death, respiratory death, cardiovascular death, chronic bronchitis, and asthma were selected as the endpoints of PM_2.5_ pollution health effects, according to an exposure–response coefficient, Poisson regression model, and health-impact-assessment-related methods (the Human Capital Approach, the Willingness to Pay Approach, and the Cost of Illness Approach), assessing the health loss and economic loss associated with PM_2.5_. The results show that the pollution of PM_2.5_ in Changsha City is serious, which has resulted in extensive health hazards and economic losses to local residents. From 2013 to 2017, when annual average PM_2.5_ concentrations fell to 10 μg/m^3^, the total annual losses from the five health-effect endpoints were $2788.41 million, $2123.18 million, $1657.29 million, $1402.90 million, and $1419.92 million, respectively. The proportion of Gross Domestic Product (GDP) in the current year was 2.69%, 1.87%, 1.34%, 1.04% and 0.93%, respectively. Furthermore, when the concentration of PM_2.5_ in Changsha City drops to the safety threshold of 10 μg/m^3^, the number of affected populations and health economic losses can far exceed the situation when it falls to 35 μg/m^3^, as stipulated by the national secondary standard. From 2013 to 2017, the total loss under the former situation was 1.48 times, 1.54 times, 1.86 times, 2.25 times, and 2.33 times that of the latter, respectively. Among them, all-cause death and cardiovascular death are the main sources of health loss. Taking 2017 as an example, when the annual average concentration dropped to 10 μg/m^3^, the health loss caused by deaths from all-cause death and cardiovascular disease was 49.16% of the total loss and 35.73%, respectively. Additionally, deaths as a result of respiratory disease, asthma, and chronic bronchitis contributed to 7.31%, 7.29%, and 0.51% of the total loss, respectively. The research results can provide a reference for the formulation of air pollution control policies based on health effects, which is of great significance for controlling air pollution and protecting people’s health.

## 1. Introduction

PM_2.5_ refers to particles with a diameter below 2.5 μm [1,2,3]. Their impact is mainly reflected in health effects on humans. First of all, larger particulate matter can only reach the throat because of the large diameter of the particles. The PM_2.5_ has a small particle size and can travel through the nasal passages along the respiratory tract to the bronchii and lungs, affecting the gas exchange process, and causing the body to suffer from respiratory diseases [4]. Secondly, because PM_2.5_ has a larger specific surface area, it can easily carry heavy metals and toxic organic substances harmful to humans, which makes PM_2.5_ more toxic, and the toxic substances carried by PM_2.5_ can be dissolved in the blood and enter the blood circulation, causing cardiovascular diseases. Prolonged exposure to high concentrations of PM_2.5_ not only causes human illnesses, but also exacerbates diseases, seriously affecting human health and reducing life span.

At the beginning of 2013, China suffered the most severe haze since observations began [5], which made PM_2.5_ begin to attract attention. That same year, China officially included PM_2.5_ in the monitoring indicators and began monitoring in key cities. In recent years, the issue of PM_2.5_ levels exceeding the standard has become more and more serious, which has aroused widespread concern in society.

In 2013, the annual average concentration of PM_2.5_ in Changsha City reached 79.1 μg/m^3^. Although the government adopted a series of policies to control the concentration of PM_2.5_, the average annual concentration of PM_2.5_ in Changsha still reached 52 μg/m^3^ in 2017. According to the secondary standard of 35 μg/m^3^ in China’s Ambient air quality standard (GB 3095-2012) and the threshold concentration of 10 μg/m^3^ delineated by the World Health Organization (WHO) [1,6], the problem of PM_2.5_ pollution in Changsha still remains severe. 

Research on the effects of atmospheric particulate matter on human health is becoming more and more popular. Among them, the six-city study of Harvard University and the cohort study of the American Cancer Society have the characteristics of large sample size and research using measured data. Moreover, the results of the analysis support the existence of a positive correlation between the long-term exposure to atmospheric particulate matter (especially fine particulate matter) and mortality; therefore, the results are the most authoritative. The former conducted a follow-up observation of 8111 adults in six cities in the United States, and found that for every 10 μg/m^3^ increase in the average annual concentration of PM_10_, the total deaths among the population increased by 8.2% [7]. The second study tracked the American Cancer Society cohort study and collected more particulate data. The survival status of 500,000 adults (≥30 years old) in large cities in the United States after exposure to different levels of air pollution was analyzed. It was found that the risk of total death in the population increased by 4.0% when the concentration of PM_2.5_ increased by 10 μg/m^3^ [8]. In addition, Beelen et al. [9,10] conducted a study of European cohorts for air pollution effects (ESCAPE) to explore the link between long-term air pollution and natural mortality. This series of studies consisted of 22 cohort studies based on European exposures and concluded that a 5 μg/m^3^ increase in PM_2.5_ concentration would increase total mortality by 7% and lung cancer by 18%.

There are also scholars in China who have conducted related cohort studies. Cao et al. studied the relationship between the deaths of 70,947 people in 31 cities in 16 provinces in China from 1991 to 2000. It was found that the total suspended particles (TSP) increased by 10 μg/m^3^, all-cause death, cardiovascular death, respiratory disease death, and lung cancer deaths increased by 0.3%, 0.9%, 0.3%, and 1.1%, respectively [11]. Zhang et al. used data for 12 years from 1998 to 2009 to study the relationship between cardiovascular and cerebrovascular death and PM_10_, SO_2_, and NO_2_ and calculated their relative risks [12]. Chung et al. [13] used a Bayesian hierarchical model to study the relationship between PM_2.5_ and mortality. The results showed that prolonged exposure to PM_2.5_ would lead to an increase in mortality. Pope III et al. [14] combined adult risk factors with air pollution data.

Under ideal conditions, the quantitative study of the effects of atmospheric particulate matter on health should be based on the exposure–response coefficient after local investigation in the study area. However, due to the difficulty in obtaining the exposure reaction coefficient, in recent years, many scholars have used relevant research results to determine the exposure–response coefficient by using meta-analysis methods. Stieb et al. conducted a comprehensive meta-analysis of 109 articles and conducted a comprehensive and systematic synthesis of daily time series studies of air pollution and mortality worldwide [15]. La et al. conducted a meta-analysis of 26 studies in 24 regions, including China, to study the relationship between NO_2_ and all-cause death, cardiovascular death, respiratory death, and cardiopulmonary death [16]. By comparing the exposure–response relationship between atmospheric particulate matter pollution and population mortality in China, Europe and the United States, Kan found that the impact of atmospheric particulate matter pollution on residents’ mortality is lower than that of developed countries. It is believed that this may be different due to the different levels of air pollution in different countries and regions, the susceptibility of the local population to air pollution, and the age distribution of the population, especially to different particulate components [17]. Aunan et al. published a report on the effects of PM_10_ and SO_2_ pollution on human health in China [18]. 

Health impact assessments rely on exposure–response coefficients. They use exposure–response coefficients to quantify the effects of contaminants on human health. The widespread use of health impact assessments in air pollution has prompted researchers to develop software tools to help assess the health effects of air quality changes. One of the best known is the Environmental Benefits Mapping and Analysis Program (BenMAP). It was developed by the US Environmental Protection Agency to implement a personalized health impact assessment (HIA) and benefit-cost analysis for air quality control. The environmental benefit assessment system has been widely used in the United States, South Korea, Spain, and other countries. Boldo et al. used BenMAP to assess the health effects of reduced PM_2.5_ in Spain [19]. Bae et al. used the BenMAP model to assess the health effects of improved aging air quality in South Korea [20]. Morefield et al. described a new modeling and analysis framework, built around the BenMAP, for estimating heat-related mortality as a function of changes in key factors that determine the health impacts of extreme heat [21]. The model is rarely used in China. Duan et al. used the BenMAP model to assess the impact of changes in PM_10_ pollution in the Pearl River Delta region on public health [22].

In addition, the most widely used health economic assessment method is the internationally accepted Poisson regression proportional hazard model. Lv et al. [23] used the Poisson regression proportional hazard model to evaluate the health economics of PM_2.5_ and PM_10_ pollution in the Beijing-Tianjin-Hebei region.

The quantitative assessment of the economic losses caused by environmental pollution has also attracted more and more scholars. Quantitative assessment of health losses caused by atmospheric pollution and monetization estimation is the basis for a cost-benefit analysis of environmental protection measures. This is of great significance to strengthening the government’s efforts in air pollution control. Three assessment methods are used to assess public health losses from air pollution: the Human Capital Approach (HCA), the Willingness to Pay (WTP), and the Cost of Illness Approach (COI). Alberini et al. used the COI and the WTP method to assess the economic loss of respiratory diseases caused by air pollution [24]. Zmirou et al. translate medical care use and absences associated with respiratory diseases into direct and indirect medical and social costs through the use of COI [25]. Xie et al. combined the exposure–response model and the COI to investigate the occurrence of heavy pollution in Beijing from January 10–15, 2013. The study found that the health and economic losses caused by the six days of smog weather in Beijing reached 489 million yuan [26].

According to the exposure–response relationship, Poisson regression model, and health impact assessment related methods (WTP, HCA, COI), this study will analyze the following issues: (1) The PM_2.5_ situation in Changsha City. (2) the health and economic losses caused by PM_2.5_ pollution in Changsha. (3) Proposed measures to reduce health and economic losses.

## 2. Materials and Methods

### 2.1. Research Area

As the capital city of Hunan Province, Changsha is located in the northern part of the province. The latitude and longitude span are between 111°53′~114°15′ east longitude and 27°51′~28°41′ north latitude. There are six districts in Changsha, namely Furong, Tianxin, Yuelu, Kaifu, Yuhua, Wangcheng (collectively known as Changsha City), and three counties, Changsha, Liuyang and Ningxiang [27].

There are 10 air quality testing stations in Changsha. One of them is a clean control point, set in a sand flat with good air quality and away from pollution sources. The other nine monitoring sites are the Economic Development Bureau of the Economic Development Zone, the Environmental Protection Bureau of Gaokai District, Mapoling, Hunan Normal University, Yuhua District Environmental Protection Bureau, Wujialing, Changsha Railway Station, Tianxin District Environmental Protection Bureau, and Hunan University of Traditional Chinese Medicine. Their specific location distribution is shown in Figure 1.

### 2.2. Data Sources

The daily concentration data from December 2013 to December 2017 are obtained from the PM_2.5_ historical data website. The number of days in which the primary pollution factor is PM_2.5_ in 2017 are obtained from the official website of the Changsha Environmental Protection Bureau. Table 1 lists the monthly PM_2.5_ concentration data statistics for 2017.

The total Gross Domestic Product (GDP) and GDP per capita data of Changsha City from 2013 to 2017 are derived from the statistical bulletin of Changsha City. The mortality rate and resident population was obtained from the Changsha Statistical Yearbook. Respiratory disease mortality was obtained from the Health and Family Planning Statistics Yearbook. The cardiovascular mortality rate was obtained from the Chinese cardiovascular disease reports over the years. The prevalence of chronic bronchitis is taken from the 2015 Chinese Nutrition and Chronic Disease Status Report. The prevalence of asthma was published by the China Asthma Alliance in 2013. 

### 2.3. Research Methods

#### 2.3.1. Meta Statistical Analysis

The exposure–response relationship is the key to quantitatively assessing the health effects of PM_2.5_ on people, linking the changes in PM_2.5_ concentration to the health-effect endpoints. Meta-analysis can comprehensively analyze the results of exposure–response coefficients of multiple literature sources and obtain the mean and 95% confidence interval.

The choice of literature follows the following principles: (1) We selected published research on the relationship between atmospheric particulate matter and human health. (2) There is a clear quantitative expression of exposure–response in the literature and, considering the age heterogeneity of the calculation of each exposure–response coefficient, the selected literature research population should be the whole population. (3) Due to the low concentration of atmospheric particulate matter in foreign countries, the exposure reaction coefficients are quite different from that in China. Therefore, the research area of the selected literature should be the literature of various regions in China which contains large amounts of detailed, high-quality information.

The meta-analysis process is as follows: First, the heterogeneity test is performed on the collated literature. Using the following two methods, (1) I^2^ test, when I^2^ < 25%, it is considered that there is no heterogeneity; when 25% < I^2^ < 50%, the degree of heterogeneity is small; when 50% < I^2^ < 75%, there is a certain heterogeneity; when I^2^ > 75%, there is a large heterogeneity. In this study, when I^2^ is less than 50%, the fixed-effect model is selected, and when it is greater than 50%, the random effect model is selected. (2) The P value is greater than 0.1, indicating that there is no heterogeneity, and the fixed-effect model is selected; if the P value is less than or equal to 0.1, it is considered to be heterogeneous, that is, there is a difference between the studies, and the random effect model is selected. Second, a meta-analysis based on the selected fixed or random effects model yields mean and 95% confidence limits. Finally, a sensitivity analysis of the health-effect endpoints calculated with the random effects model was performed to examine the effect of individual studies on overall outcomes.

#### 2.3.2. Passion Regression Model

In the context of the total population of the city, the probability of occurrence of each health-effect endpoint is low, which is a small probability event. As a time series model, the actual distribution is consistent with the statistical Poisson distribution. Therefore, most of the current health risk assessments of atmospheric particulate matter pollution are based on the relative risk model of Poisson regression [23]. That is, the health effect caused by the change in PM_2.5_ concentration is quantified by calculating the health loss caused by the increase in the PM_2.5_ concentration. The formula is as follows:
E = Exp[β*(C − C_0_)]*E_0_(1)
ΔE = E − E_0_(2)

In the above formula, E represents the population health effect under the actual concentration of PM_2.5_, E_0_ represents the population health effect under the PM_2.5_ threshold concentration, β represents the exposure–response relationship coefficient of a health-effect endpoint, and C represents the actual PM_2.5_. Concentration, C_0_ represents the threshold concentration of PM_2.5_, and the threshold concentration is 35 μg/m^3^ in China’s secondary standard and 10 μg/m^3^ in WHO. △E indicates the excess health effect, that is, the difference in health effects between the actual concentration and the threshold concentration.

#### 2.3.3. Environmental Value Evaluation Methods

##### Willingness to Pay Approach (WTP)

The willingness to pay method can more fully reflect the negative effects of the relevant health-effect endpoint on the economic losses brought by the research population and reflect the personal will and payment preferences of the research population. In general, the economic value of residents’ deaths is measured by the value of a statistical life (VOSL) [28]. The World Bank conducted a study on residents’ willingness to pay for health risks in three regions of China in 2010 including Danyang City (Jiangsu Province), Liupanshui City (Guizhou Province) and Tianjin City. The results show that the average value of VOSL in these three regions is 795,000 yuan. (approximately $115,195.5) [29]. This study used the benefit transfer method (BTM) to calculate the VOSL value of Changsha City. Due to the difference in income levels between Changsha and Tianjin, the per capita annual income ratio is used as a factor to convert, see Equation (3). For the economic loss accounting for the health-effect endpoint of death, the formula is as shown in Equation (4):(3)$$VOSL_{CS}=79.5\times {\left(\frac{I_{CS}}{I_{TJ}}\right)}^{\varphi }$$
(4)$$C_{d}=VOSL_{CS}\times \Delta E$$

*I_CS_* in the above formula represents the average per capita disposable income in Changsha from 2013 to 2017. I_TJ_ represents the average of per capita disposable income in Tianjin from 2013 to 2017. e indicates income elasticity (generally, the value is 1). *C_d_* represents a healthy economic loss.

The calculation of Changsha VOSL in the above formula uses the benefit transfer method (BTM). The difference in the income of residents in different locations is an important factor in the benefit transfer method. Therefore, the elasticity coefficient, e, is used to express the relationship between the willingness to pay and the increase in income. The change of the elastic coefficient has a great influence on the final evaluation result. When the elastic coefficient increases from 0.4 to 1.1, it will eventually differ by 20 times. Therefore, when there is no relevant data suggesting the appropriate value of the elastic coefficient, we use a higher elastic coefficient of 1.0 as the estimated willingness to pay for the health-effect endpoint [30].

All-cause death, respiratory disease deaths, and cardiovascular disease deaths are all assessments of the economic value of death. Therefore, we use Equation (3) to estimate the unit economic loss. According to the calculation, the VOSL of Changsha City is 942,100 yuan (approximately $136,510.29). Then, Equation (4) is used to calculate the economic loss of the two. The daily per capita value of Changsha’s GDP is calculated based on the annual gross domestic product of Changsha City and the corresponding number of days per year. 

##### Human Capital Approach (HCA)

The Human Capital Approach treats people as production machines and calculates the present value (discounted value) of the loss of income due to premature death. The Human Capital Approach can more realistically reflect the labor value lost by people who died prematurely due to air pollution.

The average number of lost days in a unit of chronic bronchitis refers to the results of Gu [31], that is, the average number of lost days is 1.38 days. The cost of treatment is 2000 yuan (approximately $289.8) through inquiry.

##### Cost of Illness Approach (COI)

Considering that asthma is extremely difficult to cure [32], it can only be controlled with drugs. Therefore, the health economic loss of asthma is accounted for in the form of unit case loss. The unit economic value of asthma in Beijing is 1840.3 yuan (approximately $266.66) per case [33], and the conversion to the market is 1523.3 yuan (approximately $220.73) per case. 

For the loss in economic value caused by chronic bronchitis, the cost of illness approach is adopted [34], using the following formula:
C = (C_p_ + GDP_CS_ × T) × I(5)

In the formula, C indicates the total cost of disease caused by chronic bronchitis, C_p_ indicates the treatment cost per unit of chronic bronchitis, GDP_CS_ indicates the daily per capita value of Changsha’s GDP, and T indicates the time lost due to chronic bronchitis. I indicates the amount of change in the health effects of chronic bronchitis due to PM_2.5_.

## 3. Results

### 3.1. PM_2.5_ Concentration Status of Changsha

#### 3.1.1. Monthly Trends

The variation of PM_2.5_ concentration from December 2013 to December 2017 is shown in Figure 2. It can be seen that the annual PM_2.5_ changes are characterized by a U-shaped curve, that is, the highest concentration is observed in January, followed by a downward trend, and the concentration in July and August is low, and then increases. The U-curve characteristic is most significant in 2017, with a minimum concentration of 24 μg/m^3^ in August 2017 and a maximum concentration of 99 μg/m^3^ in January. In 2014, the lowest concentration of 47 μg/m^3^ appeared in August, and the highest concentration in January was 136 μg/m^3^, with small peaks in June and October. In 2015, the lowest concentration in June was 31 μg/m^3^, the highest concentration in January was 115 μg/m^3^, and local high values appeared in May and October. In 2016, the lowest concentration of 33 μg/m^3^ appeared in June and the highest concentration in December was 84 μg/m^3^. The local high value appeared in September. In addition, it can be seen intuitively from Figure 1 that the PM_2.5_ concentration is decreasing year by year and the concentrations in January and December 2017 are higher than the corresponding months in 2016.

Since the trend in PM_2.5_ is almost the same every year, they are all U-shaped. Therefore, in order to further understand the monthly changes of PM_2.5_, we select the 2017 data for a more detailed analysis. The number of days in which the monthly primary pollution factor is PM_2.5_ in 2017 is shown in Figure 3. It can be clearly seen that for a large number of days in January and December, the primary pollution factor is PM_2.5_, and the calculated ratio is 90.32%. In August, PM_2.5_ was never the primary pollution factor. It is evident that the monthly change in PM_2.5_ as the primary pollution factor also appears as a U-shaped curve.

Since the average concentration does not reflect the extreme concentration of PM_2.5_, the daily concentration is obtained from the PM_2.5_ historical data website. The lowest concentration of 3 μg/m^3^ appeared on 4 October 2017. The highest concentration on 28 January 2017 was 284 μg/m^3^. The monthly concentration status is analyzed according to the results obtained in Figure 4. It can be clearly seen from the figure that the air quality in January is not good and that the concentration of PM_2.5_ reaches 284 μg/m^3^. Furthermore, the number of days exceeding the national secondary standard in January reached 28 days. Similarly, the air quality situation in December is not optimistic, and the concentration was mostly concentrated between 35 μg/m^3^ and 115 μg/m^3^, up to 162 μg/m^3^. In comparison, the air quality was good from June to August, and the concentration of PM_2.5_ is concentrated in the interval of 0 to 35 μg/m^3^. The lowest concentrations in these three months were 11 μg/m^3^, 9 μg/m^3^ and 10 μg/m^3^. The highest values were 57 μg/m^3^, 65 μg/m^3^ and 44 μg/m^3^. The concentration of PM_2.5_ is mostly concentrated in the interval of 0 to 75 μg/m^3^ in Changsha in 2017, and the extreme concentration is a minority occurrence.

On the whole, the higher concentration of PM_2.5_ is basically concentrated in January, November, and December during winter, after which the average concentration shows a downward trend. The lower concentration of PM_2.5_ is basically concentrated in July and August of summer, and then, the average concentration begins to increase gradually. The reason for presenting a U-shaped curve needs to be considered due to the presence of four seasons. The reasons for the higher concentration of PM_2.5_ in winter are as follows. First, the Spring Festival is generally held over the winter season. This is when the highest frequency of fireworks and firecrackers occurs. The discharge of fireworks and firecrackers is also a source of fine particles in the air. Second, the temperature in Changsha in winter is low, between 5 °C and 12 °C. Therefore, people will take measures such as burning coal to meet their own living needs and heating needs, resulting in a high concentration of PM_2.5_ in Changsha in winter [35,36]. Third, due to evident radiation radiance on the ground in winter at night, the inversion temperature is prone to occur in low altitudes and the inversion temperature is also increased. Particulate matter that is released into the atmosphere is confined in the lower atmosphere and accumulates, resulting in PM_2.5_ contamination in winter. Fourth, the northwest wind is dominant in Changsha in winter, and the concentration of PM_2.5_ is more susceptible to the input of exogenous pollution factors in the north with the wind [27]. In spring, the humidity in the air begins to gradually increase. PM_2.5_ will condense into larger droplets or form fog, which will reduce the PM_2.5_ content in the atmosphere, resulting in a significant decrease in the concentration of fine particles in the atmosphere from the onset of spring. There are three reasons for the low concentration of PM_2.5_ in summer. First, Changsha has a high temperature in summer, a large amount of rainfall and heavy rainfall, and it has a strong leaching effect on particulate matter, which can effectively remove particulate matter in the air. Second, due to the temperature in summer, inversion is not easily produced, and particulate matter does not accumulate easily. Third, Changsha is dominated by the southeast monsoon in summer, and the particulate matter blown from the north has little effect on the air quality of Changsha [37]. In autumn, due to the relatively dry weather and the burning of straw after the autumn harvest, the concentration of fine particles in the atmosphere began to increase significantly.

#### 3.1.2. Annual Trends

It can be seen from Figure 5 that the average annual concentration of PM_2.5_ in Changsha has been decreasing year by year from 79.1 μg/m^3^ in 2013, and the average annual concentration of PM_2.5_ in 2017 dropped to 52 μg/m^3^. It can be seen that the amount by which PM_2.5_ exceeds the standard in Changsha has been greatly improved. However, according to the second-class ambient air functional zone in China’s environmental quality standards, the annual average concentration of PM_2.5_ in Changsha City is still much higher than the defined concentration. In addition to this, according to the WHO’s interpretation of the particulate air quality guidelines, the human death threat will increase by about 15% at 35 μg/m^3^ compared to 10 μg/m^3^ [38]. It seems that the PM_2.5_ in Changsha is still exceeding the standard, and people’s health is still in danger.

From the perspective of annual changes, the overall trend is clearly decreasing. The reason for this is that Changsha City has formulated an annual plan for air pollution prevention and control, strengthened measures to prevent air quality deterioration, and increased the investment in related costs. During 2016, the pollution in Changsha City was comprehensively addressed; the use of highly polluting fuel in production processes was rectified and the remediation of soot was strengthened. At the same time, the government has increased certain efforts regarding strengthening the elimination of yellow-label vehicles and the current road inspection enforcement work; controlling the dust on construction sites and setting up barrier walls for the sites under construction to prevent particulate matter from entering the space outside the fence to a large extent. This series of measures has made the average annual concentration of PM_2.5_ in Changsha lower, but it is not negligible, and the average annual concentration of PM_2.5_ still reaches 52 μg/m^3^, meaning that PM_2.5_ is still the primary pollution factor. This problem requires an urgent solution.

#### 3.1.3. Comparison of Concentration in Each City in Hunan Province

Taking the average annual concentration of PM_2.5_ in 2017 as an example, the PM_2.5_ status of 14 cities and autonomous prefectures in Hunan Province was compared and analyzed, as shown in Figure 6. It can be seen from the figure that the PM_2.5_ concentrations in 14 cities and autonomous prefectures in Hunan Province in 2017 exceeded the national air quality secondary standard. This problem whereby PM_2.5_ exceeds the standard is very serious. The annual average concentration is ranked from small to large, and it can be concluded that Changsha ranks 11th among them.

The PM_2.5_ pollution in Changsha City can be analyzed from the following two aspects: From a spatial point of view, the PM_2.5_ pollution in Changsha City is serious because Changsha is located in the Xiangjiang Valley, and its overall topographical structure is a horseshoe shape with a northward opening. Before the formation of effective rainfall, the particulate matter blown from the north remained in the Changsha area and was difficult to distribute, resulting in a higher concentration of PM_2.5_ in Changsha. Judging from the development of Changsha City, as the capital city of Hunan Province, the economic development level is faster than that of the other 13 cities. The city is large in scale, densely populated, with a high level vehicle possession and large traffic volume, which will lead to a large amount of domestic pollutants and automobile exhaust emissions. In terms of industrial production, the consumption of coal in Changsha is high, resulting in a large amount of particulate matter being discharged into the atmosphere. In addition, since the fireworks display activities in Changsha Orange Island in 2010, a large amount of particulate matter generated during the fireworks display has a certain impact on the air quality of Changsha City. All of the above reasons will result in higher PM_2.5_ concentrations in Changsha compared to other cities in Hunan Province.

### 3.2. Population Health Effects

#### 3.2.1. Exposure Population

First, considering that the air quality is strongly related to the degree of industrialization of the city, the most likely population exposed to air pollution is the urban population. Second, there are 10 air quality monitoring stations in Changsha, most of which are located in urban areas. Third, the existing epidemiology usually takes the urban resident population as the main exposed population [39]. Based on the above reasons, the study selected Changsha’s “resident permanent population” as the population exposed to PM_2.5_.

#### 3.2.2. Exposure Route

There are two main ways in which the human body can be exposed to PM_2.5_, namely breathing and ingestion. The only way to have a direct effect on human health is that the human body through the respiratory action allows the fine particulate matter of the atmosphere to enter the human body through the respiratory tract and smoothly pass through the bronchii, thereby affecting gas exchange in the lungs [4]. Second, fine particles can fall into the food or water in the atmosphere. The human body can transfer fine particles of food and water into the body by feeding and drinking water, which will have certain health effects on the human body.

#### 3.2.3. Health-Effect Endpoint

It has been proven that PM_2.5_ can cause health risks to the population: damage to the lungs or other parts of the respiratory system, affecting the cardiovascular system, increasing the rate of premature death, increasing the risk of cancer, proneness to chronic bronchitis, emphysema, and asthma [17,33]. The choice of the health-effect endpoint in this study is based on two principles: First, the feasibility and accessibility of the required data, that is, the selected health-effect endpoint must have sufficient reliable data for the next study. The second is to select a healthy endpoint that has been identified in the existing study to have an exposure–response relationship with PM_2.5_. Considering the above two principles, this paper chooses the following five health-effect endpoints, including all-cause deaths, chronic bronchitis and asthma, as well as death from respiratory and cardiovascular diseases.

#### 3.2.4. Meta Statistical Analysis

Changsha has fewer health-effect endpoint studies. Therefore, this study chose to collect the literature that directly or indirectly referred to the changes in the endpoints of health effects caused by domestic PM_2.5_ pollution, extracting the exposure–response relationship coefficient (or the increase in risk) and the corresponding standard error in the literature [17]. The effects of the health effects endpoints extracted from the literature selected in this study are shown in Table 2.

Using the Stata software for meta-analysis, data from multiple studies describing the same end-of-health effect were combined to obtain a mean and 95% confidence interval for each healthy effect endpoint. The conclusions of the analysis indicate the percentage increase in various adverse health effects in the population caused by a rise of 10 μg/m^3^ of PM_2.5_.

Heterogeneity tests of I^2^ and *p* values were performed on the five health-effect endpoints. The results are shown in Table 3. The three effect endpoints of mortality passed the heterogeneity test, and the mean and 95% were calculated using the fixed-effect model. Confidence intervals; the two health-effect endpoints of morbidity did not pass the heterogeneity test. That is, the heterogeneity between the documents is large, and the random effects model is used for meta-analysis. After the sensitivity analysis of the results, the included literature was removed one by one, and the results did not reverse.

According to the results of the meta-analysis, for every 10 μg/m^3^ increase in PM_2.5_, all-cause mortality will increase by 0.377% (0.319%~0.445%), the mortality rate of respiratory diseases will increase by 0.366% (0.212%~0.631%), and the mortality rate of cardiovascular diseases will increase by 0.464% (0.384%~0.56%), the prevalence of chronic bronchitis will increase by 1.088% (1.044%~1.133%), and the prevalence of asthma will increase by 1.552% (1.331%~1.811%). It can be seen that when the concentration of PM_2.5_ is increased, it has a greater impact on chronic bronchitis and asthma.

### 3.3. Health Effect Estimation

Since the baseline incidence rate of each health-effect endpoint in Changsha City is difficult to obtain, it was replaced by the national baseline incidence rate. Respiratory disease mortality comes from the Health and Family Planning Statistics Yearbook [53]. The cardiovascular mortality rate comes from the Chinese cardiovascular disease reports over the years. Since the cardiovascular mortality rate in 2017 has not been announced, the 2017 cardiovascular mortality rate is based on 2016 data [54]. Chronic bronchitis and asthma have not published the prevalence rate every year; therefore, data from the same year are used. That is, the prevalence of chronic bronchitis comes from the Chinese Nutrition and Chronic Disease Status Report in 2015. The prevalence of asthma was published by the China Asthma Alliance in 2013. The mortality rate is from the Changsha Statistical Yearbook in 2017. From August to December 2017, the Changsha Municipal Public Security Bureau concentrated on checking the information of the unemployed persons who had died, and the unsuccessful accounts after death and before the cancellation; therefore, the mortality rate is abnormally high. Thus, the 2017 mortality rate is based on 2016 data. The number of baseline occurrences of each health-effect endpoint in Changsha City is obtained by multiplying the number of exposed populations and the population baseline rate. The baseline population for each health-effect endpoint is shown in Table 4.

It can be seen from Table 4 that the baseline number of all-cause deaths has decreased from 2013 to 2016 year by year. Since the 2017 mortality rate is based on 2016 data, and the resident population has increased in 2017, the baseline number in 2017 has increased from 2016. The number of baseline deaths from respiratory diseases showed a trend of decreasing first and then increasing. Among them, the number of baseline deaths reached a minimum in 2014. The number of deaths from cardiovascular diseases has been on the rise, and the maximum number of baselines was obtained in 2017. Since chronic bronchitis and asthma use the same year’s prevalence, no analysis is performed.

Equation (1) was used to calculate the population health-effect value E_0_ from the threshold concentration of Changsha City from 2013 to 2017 (the threshold concentration is 35 μg/m^3^ in China’s secondary standard and 10 μg/m^3^ in the WHO’s standard). The excess health effect of Changsha City was obtained using Equation (2). The results are shown in Table 5.

PM_2.5_ pollution poses a great threat to the health of residents in Changsha. According to Table 5, the total number of people affected by PM_2.5_ among Changsha residents from 2013 to 2017 is 76,091, 68,882, 48,495, 36,525, and 35,942, respectively. When the PM_2.5_ concentration drops to 35 μg/m^3^, the excess health effects of the five health-effect endpoints show a downward trend overall. From the greatest to the least number of people, they are suffering from asthma, chronic bronchitis, all-cause death, death from cardiovascular diseases, and death from respiratory diseases. The total number of affected people is decreasing year by year, and the number of people at the five health-effect endpoints is decreasing year by year.

It can be seen from Table 5 that when the annual average concentration of PM_2.5_ drops to 10 μg/m^3^, the excess health effects of the five health-effect endpoints show a downward trend overall, with a slight increase from 2016 to 2017. Although the exposed population to death from cardiovascular disease, chronic bronchitis, and asthma showed an upward trend, the average concentration of PM_2.5_ from 2013 to 2016 decreased rapidly and the annual change was large. Therefore, the affected population showed a significant decreasing trend from 2013 to 2016. From 2016 to 2017, the average concentration change of PM_2.5_ was only 1 μg/m^3^, and the number of urban permanent residents increased by about 270,000. Therefore, the number of affected people from 2016 to 2017 was slightly increased.

The people were ranked as suffering from asthma, chronic bronchitis, all-cause death, death from cardiovascular diseases, and death from respiratory diseases. The number of people affected by PM_2.5_ among Changsha residents from 2013 to 2017 was 104,265, 97,365, 82,803, 75,067, and 76,360 respectively. Compared with the excess health effect when the concentration of PM_2.5_ drops to 35 μg/m^3^, the excess health effect is 1.37, 1.41, 1.71, 2.06, and 2.12 times that of the latter.

### 3.4. Health Economic Accounting

The economic losses of each healthy endpoint are calculated by Equations (3)–(5). Five health-effect endpoints were classified by death and disease. The economic loss of the three death health-effect endpoints (all-cause death, respiratory disease death, cardiovascular disease death) was evaluated by the willingness to pay approach (WTP).l At the endpoint of the disease health effect, the economic losses caused by asthma and chronic bronchitis were evaluated by the human capital approach and the cost of illness method, respectively. The results are shown in Table 6.

According to the results of Table 6, the trends in health loss caused by the five health reaction endpoints were similar. As with the previous excess health effects, the losses were reduced from 2013 to 2016, with a slight increase from 2016 to 2017 in some healthy endpoints.

When the annual average concentration of PM_2.5_ drops to 35 μg/m^3^, the total annual losses of the five health-effect endpoints from 2013 to 2017 were $1878.5 million, $1382.92 million, $889.34 million, $623.87 million, and $610.66 million. The proportion of GDP in the current year was 1.81%, 1.22%, 0.72%, 0.46%, and 0.40%, respectively. The total loss and the proportion of losses are decreasing year by year. Taking 2017 as an example, the economic losses to various health-effect terminals as a result of PM_2.5_ pollution have different contributions to the total economic loss. All-cause deaths and cardiovascular disease were the main sources of health loss, with contribution rates of 48.46% and 35.59%, respectively. In addition, for death from respiratory diseases, asthma, and chronic bronchitis, the contribution rates were 8.23%, 7.18%, and 0.54%, respectively.

When PM_2.5_ concentrations fell to 10 μg/m^3^, the total annual losses of the five health-effect endpoints from 2013 to 2017 were $2788.41 million, $2123.18 million, $1657.29 million, $1402.90 million, and $1419.92 million. The proportion of the total GDP in those years was 2.69%, 1.87%, 1.34%, 1.04%, and 0.93%, respectively. The total loss decreased from 2013 to 2016 year by year, and the total loss in 2017 increased compared with 2016; the loss ratio decreased year by year. Taking 2017 as an example, the economic losses to various health-effect terminals as a result of PM_2.5_ have different contributions to the total economic loss. All-cause deaths and cardiovascular disease were the main sources of health loss, with contribution rates of 49.16% and 35.73%, respectively. Behind these, the death rate of respiratory diseases, asthma, and chronic bronchitis were 7.31%, 7.29%, and 0.51%, respectively.

The total health loss that can be avoided at five healthy endpoints when the PM_2.5_ concentrations were reduced to 35 μg/m^3^ and 10 μg/m^3^ from 2013 to 2017, respectively, were compared. It can be seen that the economic benefits when the concentration of the five healthy endpoints is reduced to 10 μg/m^3^ are 1.48 times, 1.54 times, 1.86 times, 2.25 times, and 2.33 times that of the former.

It can be seen that, although the proportion of health economic losses in GDP due to excessive PM_2.5_ concentration in Changsha has been decreasing year by year, it is undeniable that it still results in a large proportion of the health economic losses. In particular, the economic loss that can be avoided when the concentration of PM_2.5_ falls to the safety threshold is large. Therefore, it is necessary to propose specific measures for reducing the concentration of PM_2.5_ for Changsha City and implement them as soon as possible.

### 3.5. Measures to Reduce Losses

Although the average annual concentration of PM_2.5_ in Changsha has shown a downward trend as a whole, the trend has gradually become slower and is still far higher than the national secondary standard for ambient air quality. This has resulted in huge losses to the development of the city’s economy and seriously jeopardized the health of the residents. Therefore, PM_2.5_ prevention is imperative. This study explores the measures that should be taken to reduce the concentration of PM_2.5_ in Changsha and protect the health and safety of residents from the perspectives of government, enterprises, and residents, in order to reduce the health and economic losses caused by PM_2.5_ in Changsha.

#### 3.5.1. Government Measures to Reduce Health Economic Losses

(1) Enhance the level of industrial structure. At present, the three industrial structures of Changsha City are 3.6:47.4:49.0, but the secondary industry makes up a large proportion, which is still the main driving force for the first economic growth in Changsha. The third industry has an insufficient effect on the promotion of the economy’s first position. It is necessary to continue to comprehensively reform the polluting industries, such as cement and thermal power in the secondary industry, speed up the elimination of backward production capacity, and encourage industrial enterprises to install dust removal equipment to reduce the amount of soot emitted to the outside. To fully mobilize the economic vitality of the tertiary industry and strengthen the ability to promote economic growth, it is necessary to promote the development of an industrial structure to a high-tech and deep processing level.

(2) Strengthen pollution prevention and management. First, the government should strengthen the management of motor vehicles and conduct exhaust gas measurements on motor vehicles. Vehicles that emit pollutants that exceed national standards must be repaired or prohibited from driving on the road. Secondly, the fuel quality should be improved, and the use of clean energy sources, such as electric energy and petroleum liquefied gas, should be encouraged. Finally, vehicles should be encouraged to have exhaust gas purification equipment, thereby reducing the pollution generated by the vehicle exhaust. In addition, the control of dust, coal, and waste incineration should be strengthened. For the road surface and exposed ground dust problem, frequent sprinkling can be used to fix particles to the ground to reduce the possibility of dust. For major roads, peak traffic flows, and spring and winter seasons where particulate matter exceeds the standard, the frequency of sprinkling should be increased. Furthermore, it is necessary to vigorously promote the establishment of dust control zones in some areas and expand the area of dust pollution control areas.

#### 3.5.2. Measures for Enterprises to Reduce Health Economic Losses

(1) Promote green production processes. Green production should be considered from two aspects, one of which is the production process. In the production process, we must promote a clean and low-carbon energy structure, focus on the development of clean energy, gradually increase the proportion of liquid fuels and natural gas, and gradually increase the proportion of renewable energy sources such as solar energy, hydro energy, and nuclear energy. Furthermore, we must accelerate the development of new renewable energy technologies and industries, reduce the dependence on coal, and thus reduce particulate emissions to a certain extent. The second is the product processing process. In the process of product processing, appropriate waste treatment methods should be adopted. It should not be completely incinerated but rather should be classified for reuse if possible. The parts that are available for secondary use should be selected, and the parts that are difficult to reuse should be reasonably disposed of in combination with the waste.

(2) Innovate industrial production technology. Traditional industries still occupy a certain weight in enterprises above the scale of Changsha, and the proportion of high-input and high-emission industries is large. This requires these companies to constantly identify problems in the production process, always pay attention to innovative production processes, and use new renewable energy. Secondly, enterprises can attract talent and scientific and technological personnel through a series of measures to improve their product innovation capabilities and reduce the emission of pollutants while improving product quality.

#### 3.5.3. Individual Measures to Reduce Health and Economic Losses

(1) Establish a green travel concept. Green travel involves transportation methods that produce lower carbon emissions. In fact, this is a way of reducing fuel use while ensuring efficiency and also contributes to better human health. Green travel includes walking and public transportation, which can alleviate traffic pressure to a certain extent, and more importantly, reduce vehicle exhaust emissions, thereby reducing the concentration of PM_2.5_ in the atmosphere.

(2) Awareness of enhancing the quality of the air environment. On the one hand, we can learn about the current air quality, the corresponding PM_2.5_ concentration status and related health recommendations through TV, the Internet, and other channels. In the periods where PM_2.5_ content is high, the number of unnecessary outings should be reduced to reduce the risk of harm to our health. Secondly, in order to reduce the harm to ourselves, it is necessary not to discharge extra fine particulate matter to the outdoors, such as burning garbage or straw. The habits of not smoking indoors and opening the windows to increase ventilation should be encouraged

## 4. Discussion

In this study, a detailed analysis of PM_2.5_ health effects was carried out. In addition to the conclusion that the PM_2.5_ pollution situation in Changsha was severe, it was found that the variation of PM_2.5_ concentration was seasonal. That is, the monthly transition trend in the concentration of PM_2.5_ shows the characteristics of an approximately U-shaped curve. In general, the PM_2.5_ concentration in winter is high and the summer concentration is low. Wang [36] also reached the same conclusion when studying the atmospheric particulate pollutants in Harbin, China.

When accounting for the health economic losses affected by PM_2.5_, the choice of exposure–response coefficient and economic loss assessment model is the focus. The choice of study area and health-effect endpoints will have a large impact on the exposure–response coefficient. At present, it has been suggested that the coefficient of exposure–response of the cause of the disease is usually greater than the coefficient of exposure–response regardless of the cause [18]. Pope III et al. [14,55] studied the relationship between PM_2.5_ concentration and total mortality, cardiopulmonary disease mortality, and lung cancer mortality. Li et al. [56] estimated the mortality associated with PM_2.5_ in China’s provinces. Their data showed that 2.19 million (2013), 1.94 million (2014), and 1.65 million (2015) premature deaths were attributed to long-term exposure to PM_2.5_. The exposure–response coefficients used in this study were calculated by a meta statistical analysis using the exposure–response relationship coefficients (or increasing the risk) of the five health-effect endpoints in the literature and the corresponding standard errors.

The most common method of health economic accounting is the value of statistical life (VOSL) based on willingness to pay [57], which is common in both domestic and foreign applications [58,59,60,61]. In addition, there is the human capital approach and cost of illness approach. In this study, the endpoint of the health effect was classified according to death and disease, and the statistical life-value calculation method was used to evaluate the economic loss caused by death from respiratory diseases and cardiovascular diseases and all-cause death. The economic losses due to asthma and chronic bronchitis were assessed using the human capital approach and the cost of illness approach, respectively. The results show that the health economic loss that can be avoided by the decrease of PM_2.5_ concentration is not to be underestimated and is proportional to the decreasing concentration. The results of many scholars have also shown the same conclusion, that effective reduction of PM_2.5_ concentration can reduce health damage and obtain significant health benefits [62].

The entire research process has the following shortcomings: first, PM_2.5_ may have synergistic effects with other environmental pollution factors, such as PM_10_ and SO_2_, and this study does not take that situation into account. The results of the health effect assessment may therefore be low. Second, the health-effect endpoints only selected five health effects with sufficient data, and other health effects related to PM_2.5_ were not taken into account. The long-term bad weather causes people to feel depressed, and the negative economic losses caused by mental illness cannot be ignored. The economic losses caused by traffic accidents as a result of a poor environment also account for a large proportion. None of the above cases participated in the accounting of economic losses, but its impact could not be underestimated, which led to the calculated health effects that were lower than the real-life effects. The third is to calculate the health economic loss, referring to the results of the World Bank’s willingness to pay for residents’ health risks in three regions of China in 2010 including Danyang City (Jiangsu Province), Liupanshui City (Guizhou Province) and Tianjin City, which shows that the average value of VOSL in these three regions is 795,000 yuan (approximately $115,195.5). However, when Changsha’s VOSL value conversion was carried out, only the per capita annual income difference between the two places was considered, and the per capita annual income ratio was used for conversion. Other factors such as age composition, population ratio, and medical conditions were not considered. While WTP is related to population, economic, social, and other factors, the error of non-local data evaluation is relatively large [63], which may lead to some deviations from the actual results.

Therefore, in the future, when conducting such research, attention should be paid to finding appropriate methods to analyze the health-effect endpoints that are exposed to PM_2.5_ as comprehensively as possible. Only in this way can the results be more accurate and more realistic when estimating health economic losses and conducting health economic evaluations. In addition, in the evaluation, it is necessary to optimize the method of determining the exposure reaction coefficient and the monetization of health loss to achieve a reduction in the error.

## 5. Conclusions

This study first analyzed the PM_2.5_ data of Changsha City. Through the monthly concentration change of PM_2.5_ from December 2013 to December 2017 on a longer time scale, it can be seen that the concentration change of Changsha City is cyclical and demonstrates a decreasing trend year by year. U-curve features are presented in each cycle. Therefore, in order to further analyze its characteristics, we selected 2017 as an example to conduct a detailed analysis of the number of days in which the monthly primary pollutants are PM_2.5_ and the number of days in each concentration interval of PM_2.5_. The conclusion is drawn that the monthly change in PM_2.5_ as the primary pollution factor also shows a U-shaped curve and the concentration change has seasonal characteristics. In general, the concentration of PM_2.5_ is higher in winter, the primary pollution factor is PM_2.5_, and the proportion of the extreme concentration levels is higher. The concentration in summer is low, and the number of days in which the primary pollution factor is PM_2.5_ is small. From the perspective of the annual average concentration change, PM_2.5_ in Changsha showed a downward trend year by year, but the pollution is still serious. From a spatial perspective, there are 14 cities in Hunan Province, but the PM_2.5_ content of Changsha was higher than in 11 of them in 2017. The PM_2.5_ pollution in Changsha City is at a severe level.

In this study, the meta-analysis method was used to calculate the exposure–response coefficient, the Poisson regression model was used to obtain the health effects of PM_2.5_ pollution, and the health effects of five health-effect endpoints were evaluated by environmental value assessment methods (including WTP, HCA, COI). The health economic losses caused by PM_2.5_ pollution were estimated and assessed. The conclusions are as follows:(1)The total health effects and annual losses from the five health-effect endpoints from 2013 to 2017 are basically decreasing year by year. Taking 2017 as an example, when the annual average concentration drops to 35 μg/m^3^, the total annual loss of the five health-effect endpoints was 610.66 million dollars, accounting for 0.40% of the total GDP of that year. The economic losses caused by PM_2.5_ to various health-effect terminals have different contributions to the total economic loss. All-cause death and cardiovascular death are the main sources of health loss, with contribution rates of 48.46% and 35.59%, respectively. Behind these, for death from respiratory diseases, asthma, and chronic bronchitis, the contribution rates were 8.23%, 7.18%, and 0.54%, respectively. In 2017, when the annual average concentration dropped to 10 μg/m^3^, the total annual loss from the five health effects endpoints was $1419.92 million, accounting for 0.93% of the total GDP for that year. All-cause death and cardiovascular disease were still the main sources of health loss, with contribution rates of 49.16% and 35.73%, respectively. Behind these, for death from respiratory disease, asthma, and chronic bronchitis, the contributions were 7.31%, 7.29%, and 0.51%, respectively.(2)When the concentration of PM_2.5_ in Changsha City drops to the safety threshold of 10 μg/m^3^, the number of affected populations and health economic losses can far exceed the situation when it falls to 35 μg/m^3^, as stipulated by the national secondary standard. Taking 2017 as an example, comparing the excess health effects of five healthy endpoints when PM_2.5_ was reduced to 35 μg/m^3^ and 10 μg/m^3^, the latter was 1.37, 1.41, 1.71, 2.06, 2.12 times the former. Similarly, comparing the total health loss that can be avoided at five healthy endpoints when PM_2.5_ is reduced to 35 μg/m^3^ and 10 μg/m^3^, it can be seen that the five healthy endpoints can reduce the loss when the annual average concentration drops to 10 μg/m^3^. The former is 1.48 times, 1.54 times, 1.86 times, 2.25 times, and 2.33 times.

It is hoped that through this health economics accounting, the health effects of PM_2.5_ pollution will be quantified, which will attract the attention of the general public and relevant departments. Effective measures should be taken to control or even reduce the concentration of PM_2.5_, so as to provide a reference for regional ecological civilization construction and prompt relevant departments to improve the ecological compensation mechanism.

## Figures and Tables

**Figure 1 ijerph-16-02063-f001:**
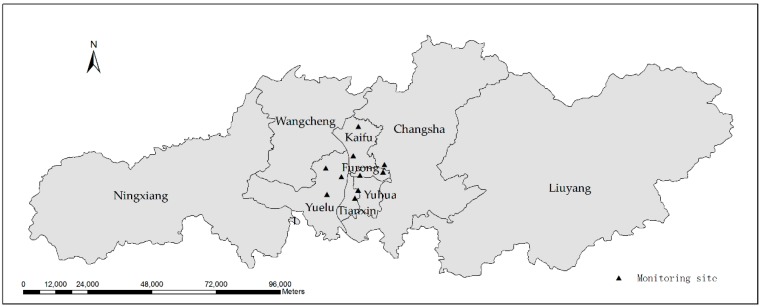
Changsha municipal administrative division and monitoring site.

**Figure 2 ijerph-16-02063-f002:**
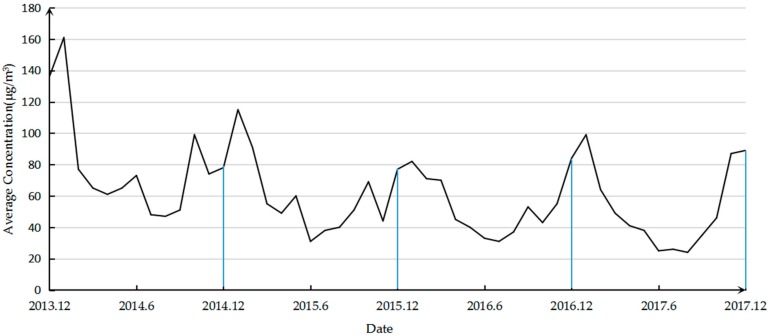
Changsha PM_2.5_ monthly variation curve from December 2013 to December 2017.

**Figure 3 ijerph-16-02063-f003:**
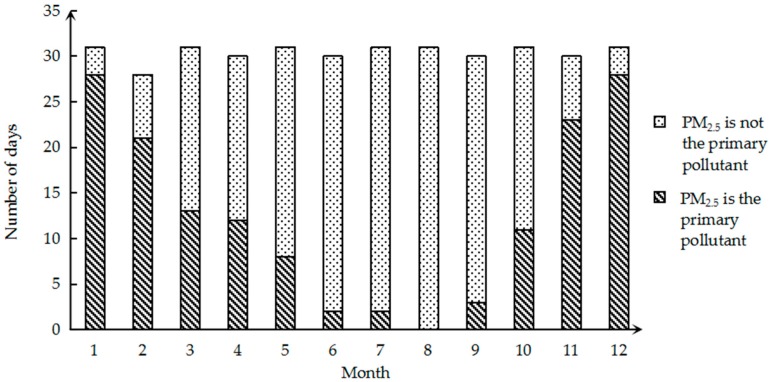
The number of days PM_2.5_ was the primary pollution factor per month in 2017.

**Figure 4 ijerph-16-02063-f004:**
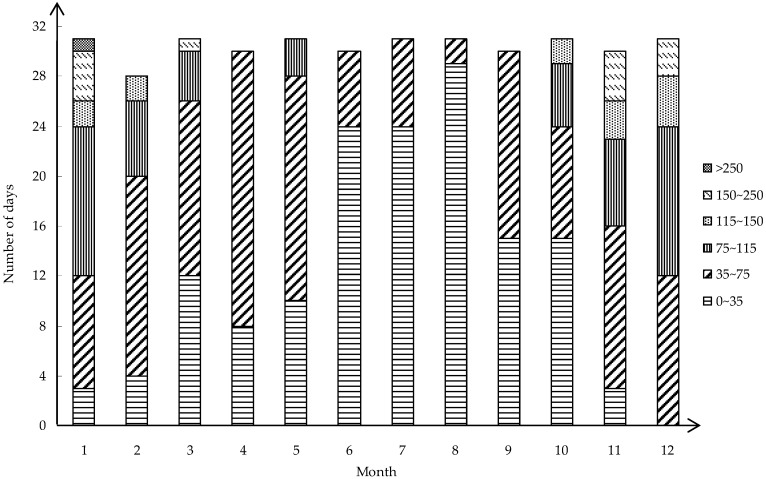
The number of days in each concentration interval of PM_2.5_ in each month of 2017.

**Figure 5 ijerph-16-02063-f005:**
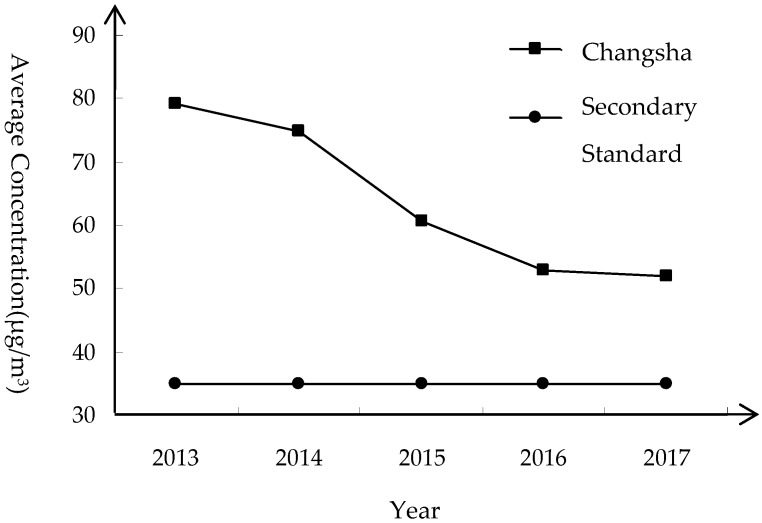
Comparison of PM_2.5_ annual average concentration and secondary standard in Changsha 2013 to 2017.

**Figure 6 ijerph-16-02063-f006:**
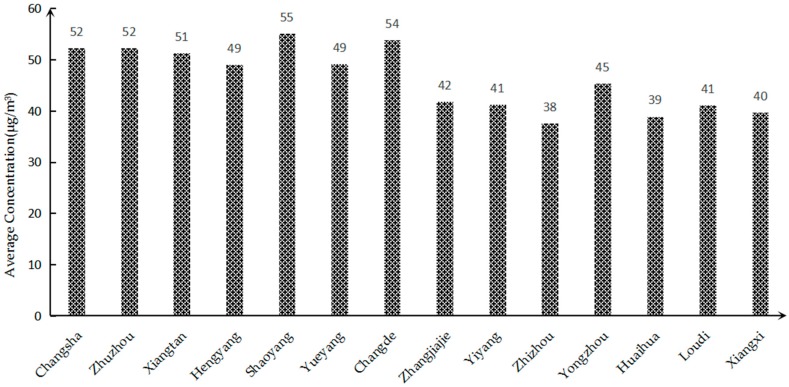
Comparison of PM_2.5_ Concentrations in Cities of Hunan Province in 2017.

**Table 1 ijerph-16-02063-t001:** Monthly PM_2.5_ concentration data statistics in 2017.

Month	Days	Average (μg/m^3^)	Max (μg/m^3^)	Min (μg/m^3^)	Number of Days PM_2.5_ Was the Primary Pollution Factor
1	31	99	284	19	28
2	28	64	133	22	21
3	31	49	173	10	13
4	30	41	65	8	12
5	31	43	79	14	8
6	30	25	57	10	2
7	31	26	65	9	2
8	31	24	44	10	0
9	30	35	69	12	3
10	31	46	135	3	11
11	30	87	182	26	23
12	31	89	162	48	28
Total	365	52	284	3	151

**Table 2 ijerph-16-02063-t002:** Exposure–response relationship for each health-effect endpoint.

Health Effects Endpoints	Exposure–Response Relationship			Author and Year
	Average (β)	Confidence Interval (95% CI)	Standard Error	
All-cause death	0.27	(0.08, 0.46)	0.10	Qian Xujun [40], 2016
	0.36	(0.11, 0.61)	0.13	Kan, H.D. [41], 2007
	0.4	(0.22, 0.59)	0.09	Feng, L. [42], 2015
	0.38	(0.31, 0.45)	0.04	Shang, Y. [43], 2013
Respiratory disease death	0.51	(0.10, 0.92)	0.21	Ge Xiyong [44], 2015
	0.63	(0.07, 1.19)	0.29	Feng Jianchun [45], 2018
	0.31	(0.10, 0.52)	0.11	Zeng Jun [46], 2017
	0.22	(0.03, 0.41)	0.10	Qi Ai [47], 2017
Cardiovascular disease death	0.285	(0.102, 0.468)	0.09	Liang Ruiming [48], 2017
	0.294	(0.041, 0.548)	0.13	Liang Ruiming [48], 2017
	0.442	(0.053, 0.832)	0.20	Liang Ruiming [48], 2017
	0.55	(0.23, 0.87)	0.16	Qian Xujun [40], 2016
	0.63	(0.35, 0.91)	0.14	Feng, L. [42], 2015
	0.44	(0.33, 0.54)	0.06	Shang, Y. [43], 2013
	0.53	(0.15, 0.9)	0.19	Xie Peng [49], 2009
	0.53	(0.09, 0.97)	0.22	Ma, Y.J. [50], 2011
Suffering from chronic bronchitis	1.09	(1.05, 1.14)	0.02	Chen Xian [51], 2016
	1.01	(0.37, 1.56)	0.33	Huang Desheng [33], 2013
	0.45	(0.13, 0.77)	0.16	Rao Li [29], 2016
Suffering from asthma	1.44	(1.36,1.52)	0.04	Chen Xian [51], 2016
	2.10	(1.45,2.74)	0.33	Xie Peng [49], 2009
	1.50	(1.2,1.7)	0.15	Fan Jingchun [52], 2016

**Table 3 ijerph-16-02063-t003:** Heterogeneity test and meta-analysis results.

Health Terminals	I^2^	*p*	Model	Average (%)	95% CI (%)
All-cause death	0	0.889	Fixed	0.377	(0.319, 0.445)
Respiratory disease death	0	0.649	Fixed	0.366	(0.212, 0.631)
Cardiovascular disease death	0	0.758	Fixed	0.464	(0.384, 0.56)
Suffering from chronic bronchitis	47.8	0.147	Random	1.088	(1.044, 1.133)
Suffering from asthma	62.6	0.069	Random	1.552	(1.331, 1.811)

**Table 4 ijerph-16-02063-t004:** Baseline number of health terminals from 2013 to 2017 (unit: person).

Year	All-Cause Death	Respiratory Disease Death	Cardiovascular Disease Death	Suffering from Chronic Bronchitis	Suffering from Asthma
2013	56,832	5532	18,732	49,106	89,545
2014	38,751	5423	19,155	49,718	90,663
2015	35,896	5452	19,682	50,536	92,154
2016	33,715	5277	20,268	51,987	94,800
2017	34,919	5321	20,992	53,843	98,184

**Table 5 ijerph-16-02063-t005:** Excess health effects when PM_2.5_ drops to 35 μg/m^3^ and 10 μg/m^3^ (unit: person).

Year	All-Cause Death		Respiratory Disease Death		Cardiovascular Disease Death		Suffering from Chronic Bronchitis		Suffering from Asthma	
	35 μg/m^3^	10 μg/m^3^	35 μg/m^3^	10 μg/m^3^	35 μg/m^3^	10 μg/m^3^	35 μg/m^3^	10 μg/m^3^	35 μg/m^3^	10 μg/m^3^
2013	8705	13,034	825	1236	3466	5138	25,952	8.94	58,905	130.02
2014	5424	8422	739	1148	3245	4987	25,206	8.79	57,602	127.14
2015	3302	6234	488	922	2204	4119	21,395	7.55	50,134	110.66
2016	2212	5046	336	769	1624	3666	19,425	6.87	46,162	101.89
2017	2168	5114	321	758	1592	3717	19,749	7.19	47,022	103.79

**Table 6 ijerph-16-02063-t006:** Value assessment of health losses when PM_2.5_ drops to 35 μg/m^3^ and 10 μg/m^3^ (unit: million dollars).

Year	All-Cause Death		Respiratory Disease Death		Cardiovascular Disease Death		Suffering from Chronic Bronchitis		Suffering from Asthma	
	35 μg/m^3^	10 μg/m^3^	35 μg/m^3^	10 μg/m^3^	35 μg/m^3^	10 μg/m^3^	35 μg/m^3^	10 μg/m^3^	35 μg/m^3^	10 μg/m^3^
2013	1188.34	1779.26	112.57	168.76	473.19	701.44	6.44	8.94	97.96	130.02
2014	740.48	1149.68	100.82	156.73	442.95	680.83	6.12	8.79	92.55	127.14
2015	450.82	851.01	66.57	125.82	300.93	562.25	4.34	7.55	66.69	110.66
2016	301.96	688.77	45.93	104.91	221.70	500.45	3.27	6.87	51.00	101.89
2017	295.92	698.05	43.82	103.50	217.35	507.40	3.31	7.19	50.26	103.79

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
