# Peer review of "Value Assessment of Health Losses Caused by PM_2.5_ in Changsha City, China"

_ijerph, 2019, doi:10.3390/ijerph16112063_

Round 1
Reviewer 1 Report
The purpose of this paper was to provide monetized value of health impacts associated with PM2.5 concentrations in Changsha City from 2013 to 2017. This study analyzed monitor data from the Changsha Environmental Protection Bureau, and using concentration-response functions from Chinese studies, estimates health burden associated with PM2.5 concentrations and the amount of burden reduced when in accordance with PM2.5 standards.
The study uses local concentration-response epidemiological functions for two mortality and two morbidity health endpoints.
The study does not provide details on the Changsha EPB monitor network, including the number of sites, dates, number of samples analyzed. The data were not filtered appropriately, as the summary information includes PM2.5 concentrations beginning at 0 ug/m3, which is not possible.
The review of literature in the introduction section does not provide sound reasoning for this study or its methods. More detail is required regarding reviewed literature methods and outcomes.
Authors present too many significant digits in all tables to appropriately convey uncertainty/error that is present in the analysis.
Population exposure could be more accurately assessed using a tool like US EPA's BenMAP-CE to relate monitored PM2.5 concentrations with local exposed population.
Throughout paper, be consistent with spaces after periods and subscripting of 2.5 within PM2.5. For all tables including confidence intervals, define the confidence interval (is it 95%? 90%?)
Page 1:
Line 12. "more and more serious". What does this mean? impacts? Quantity?
Line 21: Annual or daily concentration?
Line 31: Larger PM rather than other PM?
Line 33: Choose another word for "nuzzling"
Line 39: Reducing life span or reducing quality of life?
Line 40: Define Chinese capital - this section may not be relevant to this study/analysis.
Page 2:
Paragraph beginning at line 9: Be clear about methods - was this a source apportionment analysis? How were sources and their relative contributions confirmed?
Paragraph beginning at line 19: Update to be clear about literature and relationship between literature and this study.
Section 2.1. Provide information on PM2.5 monitoring network.
Page 3:
Line 34: add citation.
Page 5
Section 2.4, add table with the details laid out in lines 1-23.
Page 6.
Figure 1. Update Y axis to include 0.
Page 7:
Lines 1-17: Update data analysis to filter erroneous measured values (PM2.5 concentration = 0). This gives great pause to consideration of the analysis overall, as the data included in the analysis is clearly wrong. The remainder of the study will need to be updated based on appropriate inclusion of valid measurement data.
Page 10:
Update Figure 5, difficult to read. What is the purpose of this figure?
Author Response
Dear reviewer,
Thank you very much for your comments and suggestions about our paper. We have learned much from your comments, which are fair, encouraging and constructive. After carefully studying the comments and your advice, we have made corresponding changes. The cover letter with my responsese has been uploaded.
Kind regards,
Guanghui Yu

Reviewer 2 Report
Introduction: The literature review in the introduction should be reinforced. In particular, a general description of the assessment approach used in this study and an overview of the use of this assessment approach internationally.
PM2.5 concentration and national economic data: The time scale of the pollution concentration data needs to be explained. It is suggested that some descriptive statistics on the concentration of pollution of PM2.5 be presented in a table in this paragraph.
Generally speaking, the assessment approach of economic evaluation in this study is the Impact Pathway Analysis (IPA), which has been applied extensively in the measurement of damage cost. Some famous assessment frameworks internationally, such as Riskpoll (EU) and BenMAP (US) are both the applications of IPA. Therefore, I suggest that the authors add a paragraph to explain the IPA before going into the empirical analysis.
Exposure population: I belief that the exposure-response relationships you cited have their own target population with different age. Therefore, the age heterogeneity need to be considered in the selection of affected population.
Health effect estimation: In current version, authors assume that the exposure-response relationships all exist a Possion functional form. However, each health endpoint study you cited in Table 2 has their own assumption of functional form on health effect, it’s not appropriate to assume all the relationships are in line with Possion type.
In this study, the VSL is main metric and benefit transfer method (BTM) is adopted to measure the health cost of mortality in Changsha. However, benefit transfer for VSL need to take the price effect into account. In current version, only considering the income effect is not enough. Aside from that, applying BTM to VSL usually need the parameter of “income elasticity” to capture the income effect on VSL. In current version, authors give an assumption of that the elasticity value=1 without explanation. According to existing VSL studies, income effect is approved an important heterogeneity on VSL valuation. Therefore, the reason of this assumption need to be justified.
Page 5, line 16: There is a missing notation of “T” in formula (5).
The study used meta-analysis to obtain the mean of each healthy effect endpoint. More technical details about meta-analysis of this part need to be elaborated. For example, what type of mechanism you used to combine all the studies? Random effect or fixed effect? What is the assumptions and limitations of this method?
Health effect estimation: What is the reason that the excess health effects shown in Table 5 are consistently lower than that in Table6, since the air quality in Table 6 (drop to 10ug/m3) is better than that in Table 5 (drop to 35ug/m3)?
Health Economic Accounting: Theoretically, GDPcs is an indicator of economic impact, not a metric of welfare (i.e. benefit metric). In contrast, the VSL is a metric of welfare. Therefore, VSL and COI cannot be added together to become the monetization of total health effects.
Author Response

(The authors gave the same response as above.)

Reviewer 3 Report
The manuscript used to the Meta statistics analysis, Poisson regression model and Environmental value evaluation methods (including WTP, HCA, COI) to analysis variation characteristics of PM2.5 concentration in Changsha City and assessed the health and economic losses induced by PM2.5 pollutant. The authors got that a significant portion of health endpoints was related to PM2.5 concentration, and the avoidable health losses was estimated to be 0.09% and 0.21% of GDP in Changsha City when PM2.5 concentration can be controlled to 35μg/m³ and 10μg/m³, respectively. However, the language and structure of this manuscript need to be improved in order to clarify some confusions and better present the result. So, I would advise for a major revision of the manuscript before acceptance for publication. Please find my detailed commends as follows:
Comments
1. In the title section, the authors claimed to study the current situation of PM2.5, but the research period in your manuscript was from 2013 to 2017, not till now, could the authors replace “current situation”? What was health economic evaluation of PM2.5? I thought that Value assessment of Health Losses induced by PM2.5 concentration was more accurate. “Changsha” should change into “Changsha City”, “Hunan” could be deleted. Besides, your title did not seem like the title of the English article, I hope that you can refer to some relevant literatures and modify your title.
2. In the abstract, the authors mentioned that your effectively evaluated the current status of PM2.5 in Changsha City, but you simply analyzed the data of PM2.5 concentration, and you were not proposing specific measures to reduce the health economic losses based on the research results. I advised that you used an econometric model to analyze the causes of PM2.5 from the perspective of development, the number of motor vehicles and residents' consumption etc. in the purpose to enhance the depth of the manuscript. In general, I thought the part of your abstract was not highlighted and there were language problems. I suggested you rewriter the abstract.
3. In the introduction, you used two paragraphs to introduce the background information and one paragraph to summarize the research progress in China and other countries, leading a result that your literature review was not deep enough. Many classic studies related to the content of your manuscript were not cited, and literature review was not just a list, it was necessary to make your own review. I hope that you could rewrite the introduction.
4. For materials and methods, I advised that you should write it in three aspects: research area, data sources and research methods. It was best to provide a location map for the study area, indicating the administrative divisions and the location of the air monitoring stations, and did not need too much text introduction. Otherwise, your manuscript had too many tables, and many tables could be combined. Table 1 could be deleted or placed with tables 7 and 8. Data source was not reasonable enough, the raster data of PM2.5 concentration could be considered. There were also problems with the research method, I suggest you rewrite it from three aspects: Meta statistics analysis, Poisson regression model and Environmental value evaluation methods. Table 2 could be put in section 3.2. What was the threshold concentration C0 that you set in formula 1?
5. Most of the literatures you selected in table 2 were not epidemiological studies, but the results of the literatures summarized by others, which would lead a huge discrepancy when you used the Meta-analysis method to analysis these literatures. When you got the table 5, you should give the selection principle about literatures, processing steps, heterogeneity test results and literature consolidation results.
6. The ¥were used in the manuscript, for the consistency and also convenient of international readers, I would suggest the author to convert ¥to $ or make a brackets equivalent, e.g.
7. In the section 2.4, you calculated the value of health economic losses caused by PM2.5 concentration based on the survey result implemented by the World Bank in 2010, leading a fact that your workload was obviously inadequate. You might try to account the value of health economic losses by using multiple environmental valuation methods.
8. In the section 3.1, you only studied the variation characteristics of PM2.5 concentration in 2017, which seemed to lack convincing because you used one year of data to reflect the changing characteristics of the whole city. I suggested you supplement the time series and compare the spatial differences within the Changsha City. It did not seem to make much sense to compare the PM2.5 concentration for different capital cities due to huge differences in population, economy, energy and technology, etc. We suggested that you quantitatively detected the cause of PM2.5 in Changsha City.
9. For the section 3.3, the four health effect endpoints you choose were not enough (You could refer to the literature). I suggested you focused on the pathogenic or fatal effects of PM2.5 pollutant.
10. In the discussion section, the authors mentioned that the economic loss due to asthma and chronic bronchitis was assessed using the unit case loss method and the disease cost method, respectively. Please confirm the correctness of professional terms. You could easily avoid the shortcomings you pointed out, but you did not.
11. In the conclusion part, the authors mentioned that this study analyzes PM2.5 data from 2013 to 2017 in Changsha from both time and space. However, you were more focused on analyzing the temporal variation characteristics of PM2.5 concentration, and paid insufficient attention to the spatial differentiation of PM2.5 concentration. You written that from a spatial perspective, there are 14 cities in Hunan Province, but the PM2.5 content of Changsha is higher than 11 of them in 2017. I could not find it from the body of the article. The exposure-response model should change into “exposure-response coefficient”, Poisson's formula should change into “Poisson regression model”, disease cost method should change into cost of illness approach. Your conclusions were vague and superficial, and you needed to rewrite it.
12. There were serious problems with your language and structure in your manuscript. Please improve them.
Author Response

(The authors gave the same response as above.)

Round 2
Reviewer 1 Report
Most comments are addressed, I believe it can be accepted to publish at current form.
Reviewer 2 Report
Authors have revised the paper based on the comments, it can be accepted.
Reviewer 3 Report
The author has revised the manuscript according to the review opinions. I think this article has met the quality requirements of IJERPH journal, and I recommend it to be published in this journal.